# Synthesis and Property Examination of Er_2_FeSbO_7_/BiTiSbO_6_ Heterojunction Composite Catalyst and Light-Catalyzed Retrogradation of Enrofloxacin in Pharmaceutical Waste Water under Visible Light Irradiation

**DOI:** 10.3390/ma15175906

**Published:** 2022-08-26

**Authors:** Jingfei Luan, Wenlu Liu, Ye Yao, Bingbing Ma, Bowen Niu, Guangmin Yang, Zhijie Wei

**Affiliations:** 1School of Physics, Changchun Normal University, Changchun 130032, China; 2State Key Laboratory of Pollution Control and Resource Reuse, School of the Environment, Nanjing University, Nanjing 210093, China

**Keywords:** Er_2_FeSbO_7_, Er_2_FeSbO_7_/BiTiSbO_6_ heterojunction photocatalyst, enrofloxacin, pharmaceutical wastewater, visible light irradiation, photocatalytic activity

## Abstract

A new photocatalyst, Er_2_FeSbO_7_, was prepared by solid phase sintering using the high-temperature synthesis method for the first time in this paper. Er_2_FeSbO_7_/BiTiSbO_6_ heterojunction (EBH) catalyst was prepared by the solvent thermal method for the first time. Er_2_FeSbO_7_ compound crystallized in the pyrochlore-type architecture and cubelike crystal system; the interspace group of Er_2_FeSbO_7_ was *Fd3m* and the crystal cellular parameter a of Er_2_FeSbO_7_ was 10.179902 Å. The band gap (BDG) width of Er_2_FeSbO_7_ was 1.88 eV. After visible light irradiation of 150 minutes (VLGI-150min) with EBH as a photocatalyst, the removal rate (RR) of enrofloxacin (ENR) concentration was 99.16%, and the total organic carbon (TOC) concentration RR was 94.96%. The power mechanics invariable k toward ENR consistency and visible light irradiation (VLGI) time with EBH as a photocatalyzer attained 0.02296 min^−1^. The power mechanics invariable k which was involved with TOC attained 0.01535 min^−1^. The experimental results showed that the photocatalytic degradation (PCD) of ENR within pharmaceutical waste water with EBH as a photocatalyzer under VLGI was in keeping with the single-order reactivity power mechanics. The RR of ENR with EBH as a photocatalyzer was 1.151 times, 1.269 times or 2.524 times that with Er_2_FeSbO_7_ as a photocatalyst, BiTiSbO_6_ as a photocatalyst, or N-doping TiO_2_ (N-TO) as a photocatalyst after VLGI-150min. The photocatalytic activity, which ranged from high to low among above four photocatalysts, was as follows: EBHP > Er_2_FeSbO_7_ > BiTiSbO_6_ > N-TO. After VLGI-150min toward three periods of the project with EBH as a photocatalyst, the RR of ENR attained 98.00%, 96.76% and 95.60%. The results showed that the stability of EBH was very high. With appending trapping agent, it could be proved that the oxidative capability for degrading ENR, which ranged from strong to weak among three oxidic radicals, was as follows: superoxide anion > hydroxyl radicals (HRS) > holes. This work provides a scientific basis for the research and oriented leader development of efficient heterojunction catalysts.

## 1. Introduction

Antibiotics were found to be stubborn pollutants in pharmaceutical wastewater and could not be completely eliminated after conventional treatment [1,2,3,4], such as biodegradation, electrochemical process, adsorption method and flocculation settling method [5]. Enrofloxacin (ENR) was one of the common antibiotics which could effectively kill Gram-positive and Gram-negative bacteria and mycoplasma and be widely used to cure various diseases in animals [6,7,8,9,10]. However, a large amount of ENR in wastewater could promote and spread antibiotic resistance and had a negative impact on biota. Therefore, an efficient method for degrading ENR was urgently needed [11,12,13,14].

In recent years, various methods which could be used to degrade aqueous pollutants (especially microbiotic and organic dyestuff) had been developed, including chemical oxidation, biotechnology, electrochemical treatment, photodegradation and advanced oxidation. Due to the low cost and environmental benefits, photodegradation by photocatalyst was considered to be a promising technology. Therefore, many researchers focused on the design and synthesis of high-efficiency photocatalysts, which could convert the inexhaustible solar energy into the driving force of photodegradation. During the photodegradation process, antibiotics were effectively oxidized by superoxide radicals, hydroxyl radicals (HRS) or holes which derived from separated photoinduced electrons and photoinduced holes. This made the suitable band structure and effective electron hole separation particularly important for the photodegradation of antibiotics. Among various strategies (such as defect engineering, morphological variation element doping and heterostructure construction), heterogeneous junction engineering was considered to be the most promising method for achieving appropriate conduction band, valence band and charge separation capability at the same time. In particular, the heterostructure scheme system which was inspired by natural photosynthesis retained high redox capacity and was conducive to the formation of holes and free radicals [15].

Photocatalytic technology had developed rapidly since Fujishima and Honda discovered photocatalytic reaction in 1972, which attracted the attention of a large number of scholars in the scientific community. In 1976, Carey et al. studied the photocatalytic oxidation of PCBs and extended the photocatalytic technology to the field of eliminating environmental pollution. Since then, various new semiconductor materials, such as SrTiO_3_, BiVO_4_, Ag_3_PO_4_, TaON, Ta_3_N_5_, g-C_3_N_4_, CdS, MoS_2_, and their nanoparticles, had been directly applied for exploiting solar energy by various photocatalytic reactions (PLR) [16].

However, most of the traditional photocatalytic materials owned problems, such as low quantum efficiency [17,18,19,20,21], low visible light (VL) utilization and poor stability, which greatly restricted their further development. For example, commercial TiO_2_ could only absorb ultraviolet light (UV-light), which accounted for 4% of the solar energy. Therefore, light energy could not be fully utilized [22]. For the purpose of improving the catalytic activity of photocatalysts, the ion doping method, photosensitization and heterojunction construction were proved to be very effective [23,24,25,26,27,28,29]. It was well known that small changes in the inner structure of semi-conductor photocatalysts might promote the separation efficiency of photogenerated electrons and photogenerated holes, so as to increase the photocatalytic activity. Many scientists had proven that the photocatalysts with heterojunction construction possessed preferable light utilization efficiency, a long carrier life, high photocatalytic performance and high chemical stability [30,31,32,33,34,35,36]. Among the photocatalysts with heterojunction construction, p-n heterojunction architecture had been proven by many scholars to be a valid method for separating electron and hole pairs [37]. Swain et al. fabricated a AuS_2_/ZnIn_2_S_4_ p-n heterojunction photocatalyst which was prepared for the degradation of phenol in industrial sewage. From the experimental results, the degradation rate of phenol over AuS_2_/ZnIn_2_S_4_ heterojunction photocatalyst was about 15 times higher than those over pure ZnIn_2_S_4_ [38]. Liao and his colleagues constructed VL-driven BiFeO_3_/TiO_2_ p-n heterojunction composites by the simple hydrolysis precipitation method. Moreover, BiFeO_3_/TiO_2_ improved photocatalytic property owing to the ferroelectric effect of BiFeO_3_ and the effect of the internal electric field of the BiFeO_3_/TiO_2_ p-n heterojunction. As a result, the recombination of carriers was obviously inhibited, and above factors greatly promoted the degradation rate of tetracycline under visible light irradiation (VLGI) [39].

According to the previous reports, series A_2_B_2_O_7_ compounds could become better photocatalysts [40,41,42,43,44]. For example, some scientists prepared Bi_2_MNbO_7_ (M = Al^3+^, In^3+^, Fe^3+^, Sm^3+^) by the sol-gel route and the results displayed that the Bi_2_FeNbO_7_ compound family possessed momentous light-catalyzed liveness. Bi_2_FeNbO_7_, which was prepared by the sol-gel method (400 °C), presented the high light-catalyzed liveness for the degradation of methylene blue in dye waste water [45].

In our previous work [46,47,48], we found that Sm_2_FeSbO_7_ had a pyrochlore architecture. As a photocatalyzer under the condition of VLGI, the anatomical prettification of Sm_2_FeSbO_7_ seemed to have the potential for increasing the light-catalyzed liveness. In the light of above analyses, we could presume that the superseding of Sm^3+^ by Er^3+^ in Sm_2_FeSbO_7_ could improve the carriers’ concentration [49]. The results showed that the electronical transmittal and optical physics characteristics of the late model Er_2_FeSbO_7_ compound was changed and improved and might have advanced photocatalytic properties. Moreover, the catalyst material which was prepared in this study was a new compound which had not been reported. In addition, a series of experiments with Er_2_FeSbO_7_/BiTiSbO_6_ heterojunction (EBH) as a photocatalyst for the degradation of ENR were designed and compared with the previously reported composites for the degradation of ENR, reflecting the novelty of EBH. Li et al. studied the synthesis of NPG/Bi_5_O_7_I composites by the simple solvothermal method using ionic liquid 1-ethyl-3-methylimidazole iodide ([EMIM] I). The results showed that the introduction of NPG significantly improved the VL absorption capacity of Bi_5_O_7_I and the separation rate efficiency of photoinduced electrons and photoinduced holes in Bi_5_O_7_I, simultaneously. The NPG/Bi_5_O_7_I samples exhibited high photocatalytic activity for the degradation of colorless antibiotics tetracycline and ENR under VL irradiation [50]. In addition, Chen et al. found that graphene-like boron nitride, which was modified by bismuth phosphate material, could be used for promoting the photocatalytic degradation rate of ENR. The experimental results showed that under sunlight irradiation of 120 minutes, ENR was photodegraded by BiPO_4_, while the removal rate of ENR was 79.2%, 91.5% or 83.8% by 0.5 wt% BN/BiPO_4_, 1 wt% BN/BiPO_4_ or 2 wt% BN/BiPO_4_, respectively. It could be found that 1 wt% graphene-like BN/BiPO_4_ had the highest degradation rate of ENR [51]. However, although this improved the degradation rate of ENR, the results were not ideal. Therefore, the new catalyst material which was studied in this paper had a higher degradation efficiency in the degradation process of ENR, which made its work more meaningful, reflecting the advantages of the new material.

In this article, X-ray diffractometer (XRD), scanning electron microscope-X ray energy dispersive spectra (SEM-EDS), X-ray photoelectron spectrograph (XPS), synchrotron-based ultraviolet photoelectron spectroscope (UPS) and UV-Vis diffuse reflectance spectrophotometer (DRS) were used for resolving the anatomical characteristics of pure phase Er_2_FeSbO_7_ and pure phase BiTiSbO_6_, which were prepared by power mechanics controlment quomodo and elevated temperature solid phase fritting quomodo. Furthermore, the removal rate (RR) of ENR under VLGI with pure phase Er_2_FeSbO_7_, BiTiSbO_6_, N-doping TiO_2_ (N-TO) or EBH as a photocatalyst was detected. In this investigation, our aim was to prepare new-type heterogenous junction (HJ) catalysts which could remove ENR within pharmaceutic waste water under VLGI. Since the energy gap (ENG) of Er_2_FeSbO_7_ was 1.88 eV, which was less than the energy of incoming VL, it was likely to segregate the photo-induced electrons and photo-induced holes which were efficiently disjunct and difficult to be recombined. Therefore, Er_2_FeSbO_7_ was a VL responsive-type catalyst with high light-catalyzed liveness. Meanwhile, the conduction band (CB) electric potential of Er_2_FeSbO_7_ was −0.71 eV, which was more subtractive than −0.33 eV. Thus, the photo-induced electrons on the CB of Er_2_FeSbO_7_ were more easily to interreact with dissolved oxygen (DO) in water for producing superoxide anion. Superoxide anion had a powerful oxidated impact and could oxidate ENR firsthand. The valence band (VB) electric potential of BiTiSbO_6_ was 2.995 eV, which was more nonnegative than (2.38) eV. Therefore, holes in the VB of BiTiSbO_6_ could oxidate H_2_O or OH^-^ into HRS for degrading ENR in water. HRS had a powerful oxidated impact, whereupon the contamination of ENR in water could be immediately and efficiently oxidated by HRS. At the same time, the ENG of BiTiSbO_6_ was 2.32 eV. Thus, BiTiSbO_6_ was also a VL responsive-type catalyst with high light-catalyzed liveness. Both Er_2_FeSbO_7_ and BiTiSbO_6_ could produce photo-induced electrons and photo-induced cavities, which were difficult to be recombined under VLGI, and owned powerful light-catalyzed liveness. Thus, Er_2_FeSbO_7_ and BiTiSbO_6_ could structure an ideal HJ in conjunction. 

The innovation of our work was that a new type of Er_2_FeSbO_7_ catalyst and Er_2_FeSbO_7_/BiTiSbO_6_ heterojunction photocatalyst (EBHP) were synthesized by solid phase fritting using the elevated temperature synthesis method for the first time. Further, we showed that Er_2_FeSbO_7_ and EBHP were VL responsive photocatalysts with higher photocatalytic activity for removing ENR effectively. The removal of organic pollutants within pharmaceutic wastewater by using EBHP was more efficient and safer.

## 2. Result and Discussion

### 2.1. X-ray Diffractometer (XRD) Analysis

The architecture of the Er_2_FeSbO_7_ results, which were examined by XRD, are shown in Figure 1. Materials Studio (MS) software was applied to acquire the quantitative data, which took Rietveld analysis quomodo as the basis. The results show that Er_2_FeSbO_7_ was single phase and the lattice parameters of the new-type photocatalyst Er_2_FeSbO_7_ were 10.179902 Å. At the same time, the ultimate refinement for Er_2_FeSbO_7_ showed that there was good consistency between the observed strength and the surveyed strength of the pyrochlore structure, which was a cubelike crystal system with a interspace group of *Fd3m* (O atoms were included in the mold). Moreover, the total XRD data of Er_2_FeSbO_7_ could be successfully indexed to space group *Fd3m*, according to the lattice constant. Table 1 shows the atomistic co-ordinates and architecture parameters of Er_2_FeSbO_7_. Appendix A shows the atomistic architecture of Er_2_FeSbO_7_. The conclusion can be drawn from Figure 1 that Er_2_FeSbO_7_ crystallized, and the pyrochlore-type architecture was achieved. Based on the results of refinement, the R_P_ = 15.33% with space group *Fd3m* was achieved.

Be aware that the x co-ordinate of the O (1) atom could be regarded to be an index mark of the variance of the transistorization architecture on the pyrochlore-type A_2_B_2_O_7_ compounds (cubic, space group *Fd3m*), and was equivalent to 0.375 when the six A-O (1) bond distances were equal to that of the two A-O (2) bond distances [52]. Therefore, information on the deformation of the MO_6_ (M = Fe^3+^ and Sb^5+^) octahedra could be obtained from the x value [50]. The x value was shifted off x = 0.375 [50], meaning that the deformation of the MO_6_ (M = Fe^3+^ and Sb^5+^) octahedra presented visibly in the crystalloid architecture of Er_2_FeSbO_7_. Electric charge disassociation was demanded for the photocatalytic degradation (PCD) of ENR under VLGI with the view of averting the regrouping of the photo-induced sigma electrons and photo-induced cavities. Inoue [53] and Kudo [54] showed that the partial deformation of the MO_6_ octahedra, which was gained from several photocatalyzers—for instance, BaTi_4_O_9_ and Sr_2_M_2_0_7_ (M = Nb^5+^ and Ta^5+^)—was highly momentous for averting the electric charge regrouping, and contributed to increasing the photocatalytic activity. Therefore, the distortion of the MO_6_ (M = Fe^3+^ and Sb^5+^) octahedra in the crystal structure of Er_2_FeSbO_7_ could also be considered effectual for enhancing the photocatalytic activity. Er_2_FeSbO_7_ included a tri-dimensional (3D) meshwork architecture of corner-sharing MO_6_ (M = Fe^3+^ and Sb^5+^) octahedra. The MO_6_ (M = Fe^3+^ and Sb^5+^) octahedra were annexed into catena by an Er^3+^ ionized atom. Two kinds of Er-O bond distances coexisted: the six Er-O (1) bond distances 2.712 Å were visibly longer than that of the two Er-O (2) bond distances 2.295 Å. The six M-O (1) (M = Fe^3+^ and Sb^5+^) bond distances were 1.99649 Å and the M-O (2) bond distances were 4.359 Å. The M-O-M (M = Fe^3+^ and Sb^5+^) bond angles (BDAs) were 139.624° in the crystal structure of Er_2_FeSbO_7_. The Er-M-Er (M = Fe^3+^ and Sb^5+^) bond angles (BDA) were 135° in the crystal structure of Er_2_FeSbO_7_. The Er-M-O (M = Fe^3+^ and Sb^5+^) BDA were 135.505° in the crystal structure of Er_2_FeSbO_7_. The research on its luminescence characteristics results showed that if the M-O-M BDA was on the verge of 180°, then the current carrier would move with ease [52]. It revealed that the included angles were between the public angle MO_6_ (M = Fe^3+^ and Sb^5+^) octahedra. For instance, the M-O-M BDA for Er_2_FeSbO_7_ were significant for affecting the light-catalyzed liveness of Er_2_FeSbO_7_. If the M-O-M BDA was close to 180 degrees, the mobility of the photo-induced electrons and photo-induced cavities would be great [52]. The mobilities of the photoinduced electrons and photoinduced holes affected the probability of electrons and cavities for reaching reaction sites on the catalyzer skin layer, which ultimately led to the influence on the photocatalytic activity [52].

In addition, the Sb–O–Sb BDA of Er_2_FeSbO_7_ was larger, thus, the photocatalytic activity of catalyst Er_2_FeSbO_7_ was improved. As for Er_2_FeSbO_7_, Er is a 4p-block rare-earth (RE) metallic element, Fe is a 4f-block metallic element and Sb is a 5p-block metallic element. Based on the above analysis, the impact of retrograding ENR under VLGI with Er_2_FeSbO_7_ as a photocatalyst could be credited chiefly to its crystal structure and electronic structure.

Appendix A shows the X-ray diffraction pattern of BiTiSbO_6_. We marked the singlehanded diffractive apices in Appendix A. The architecture of BiTiSbO_6_ was trialed by the X-ray diffraction technique. MS software was used for obtaining the achieved quantitative data based on a Rietveld analysis quomodo. The verdict could be made that BiTiSbO_6_ was single-phase and the lattice parameters of BiTiSbO_6_ were 9.87708 Å. In the light of the refinement result, we proved that BiTiSbO_6_ crystallized with a tetragonal spinel structure and space group *I41/A*. The BDG of BiTiSbO_6_ was calculated to be 2.32 eV.

Figure 2 shows the XRD diffraction pattern of Er_2_FeSbO_7_/BiTiSbO_6_ HJ photocatalyzer before the reaction. From Figure 2, it can be seen that the pure single crystal photocatalyzer Er_2_FeSbO_7_ and single-phase photocatalyzer BiTiSbO_6_ existed. Every diffraction maximum of Er_2_FeSbO_7_ and every diffraction maximum of BiTiSbO_6_ were resoundingly labeled, and impurity was otherwise not discovered in Figure 2. Figure 2 also shows that the XRD image of the HJ Er_2_FeSbO_7_/BiTiSbO_6_ was a single-phase with perfect crystallinity, and the defects were not found inside the crystal. Thus, the photoinduced electrons and photoinduced holes which were generated during the reaction would not aggregate and compound at the defects inside the crystal catalyst, therefore, prolonging the service life of photo-induced sigma electrons and photo-induced cavities and improving the catalytic activity of the photocatalyst.

Appendix A shows the XRD diffraction pattern of Er_2_FeSbO_7_/BiTiSbO_6_ HJ photocatalyst after the reaction. A comparison of Figure 2 and Appendix A found that the XRD images of the HJ Er_2_FeSbO_7_/BiTiSbO_6_ before the reaction and after the reaction were consistent and unchanged. As a result, the HJ Er_2_FeSbO_7_/BiTiSbO_6_ had good stability.

### 2.2. UV-Vis Diffuse Reflectance Spectra

The absorption spectra of the Er_2_FeSbO_7_ sample are listed in Figure 3a,b. The absorption edge (ABE) of this new photocatalyst Er_2_FeSbO_7_ was discovered to be at 458 nm, which was within the visual range of the optical spectrum. The band gap of energy (B-GE) of the crystal semi-conductors could be certain by the point of crossing between light quantum energy capacity *hν* axes and the line calculated from the linear section of the ABE, also known as the Kubelka–Munk function (1) (called re-emission function) [55,56].
(1)$$\frac{{\left[1-R_{d}\left(h\nu \right)\right]}^{2}}{2R_{d}\left(h\nu \right)}=\frac{\alpha \left(h\nu \right)}{S}$$
where S was the dispersion facient, R_d_ was the spread reflection and α was the delegated absorbability modulus of radialization.

The optic absorbability near the energy band edge of the crystalline semiconductor conforms to Equation (2) [57,58]:*α**hν* = A (*hν* − *E*_g_)*^n^*(2)

Here, A, *α*, *Eg* and *ν* represent the ratable numeric constant, absorbability modulus, BDG and light frequency, separately. In this equation, *n* determines the properties of transitions in semiconductors. *Eg* and *n* could be reckoned by the lower row process: (1) draw ln (α*hν*) versus ln (*hν − Eg*), supposing a hemiidentic number of *Eg*; (2) infer the number of *n* tasks based on the rate of the grade in the Figure; (3) by drawing (*αhν*)^1/n^ and *hν* to refine the number of *Eg* and extrapolating the graph to (*αhν*)^1/n^ = 0. According to the above method, the *Eg* value of Er_2_FeSbO_7_ was calculated to be 1.88 eV. The estimated value of *n* was about 2 and obliquely licensed the optical transition of Er_2_FeSbO_7_.

The B-GE of Er_2_FeSbO_7_ was 1.88 eV, the B-GE of Bi_3_O_5_I_2_ was 2.02 eV [55], and the B-GE of co-doping ZnO was 2.39 eV. All the B-GE of these three chemical compounds (CMPDs) were less than 2.78 eV, which implied that above three catalyzers had VL-responsive properties and held distorted latent energy for displaying high light-catalyzed liveness under VLGI.

Appendix A show the UV-Vis diffuse reflectance spectrum of BiTiSbO_6_. In light of the above sequences and Appendix A, the number of *Eg* for BiTiSbO_6_ was calculated as 2.32 eV. The cursory value of n was about 2, which indicated discretely indirect optical lambda transition of BiTiSbO_6_.

The absorption spectra of the EBH specimen are ranked in Figure 4a,b. The ABE of this new photocatalyzer EBH was discovered to be at 620 nm, which was in the VL range of the sunlight spectra. The ABE of the photocatalyst Er_2_FeSbO_7_ was at 640 nm and that of the photocatalyst BiTiSbO_6_ was at 534 nm, which was located in the VL area of the sunlight optical spectrum. The B-GE of Er_2_FeSbO_7_ was 1.88 eV, the B-GE of BiTiSbO_6_ was 2.32 eV and the B-GE of EBH was 1.99 eV. Every B-GE of the above three CMPDs was less than 2.32 eV, which implies that the above three catalyzers had VL-range characteristics and possessed gigantic latent force for displaying high light-catalyzed liveness under VLI.

### 2.3. Property Characterization of Er_2_FeSbO_7_/BiTiSbO_6_ Heterojunction Photocatalyst

For the purpose of obtaining the skin layer chemical constitution and the quantivalency ungerade states of each constituent of EBHP, the X-ray photoelectron spectrum (XPS) was executed. Figure 5 shows the XPS investigation spectra of EBHP. Figure 6 shows the XPS spectrogram of O^2−^, Er^3+^, Fe^3+^, Bi^3+^, Ti^4+^ and Sb^5+^ which derived from EBHP. In light of the XPS full spectrogram, which is shown in Figure 5, the compound EBHP subsumed the elementary substance of Er, Fe, Bi, Ti, Sb and O. In light of the XPS analysis effects, which are exhibited in Figure 5 and Figure 6, the oxidation state of Er, Fe, Bi, Ti, Sb or O ion was +3, +3, +3, +4, +5, or −2, separately. Due to the above analysis effects, it could be summarized that the chemical formula of the new CMPD was Er_2_FeSbO_7_/BiTiSbO_6_. It can be seen from Figure 6 that multifarious elemental apices with unique binding energies were attained. In Figure 6, the O1s peak of O elementary substance is situated at 530.54 eV; the Er_4d_ and Er_4d5/2_ peak of Er elementary substance is situated at 164.47 eV and 159.08eV; the Fe_2p3/2_ peak of Fe elementary substance is situated at 712.40 eV; the Sb_3d3/2_ and Sb_3d5/2_ peak of Sd elementary substance is situated at 539.99 eV and 530.59 eV; the Bi4f_5__/2_ and Bi_4f7__/2_ apices of Bi elementary substance are situated at 164.39 eV and 159.03 eV; the Ti_2p1/2_ and Ti_2p3/2_ peak of Ti elementary substance are situated at 465.12 eV and 458.18 eV; the Sb_4d_ peak of Sb elementary substance are situated at 35.01 eV. In a word, Figure 5 and Figure 6 exposited the existence of erbium (Er_4d_); ferrum (Fe_2p_); antimony (Sb_3d_ and Sb_4d_); bismuth (Bi_4f_); titanium (Ti_2p_); and oxygen (O1s) within the prepared specimen. The skin layer elemental analysis effects exhibited that the media atomic rate of Er:Fe:Sb:Bi:Ti: O was 309:164:404:201:208:8714. The atomic ratio of Er:Fe:Sb and Bi:Ti:Sb in the sample of EBHP was 1.88:1.00:1.13 and 0.92:0.95:1.00, respectively.

The reason for the high oxygen value is that there was a lot of adsorbed oxygen on the skin layer of EBHP. Obviously, neither shoulders nor widening in the XPS apices of EBHP was observed, meaning that the as-prepared compound was pure phase.

Figure 7 shows the XPS survey spectrum of the EBHP after the reaction. Figure 8 shows the XPS spectra of Er^3+^, Bi^3+^, O^2−^, Sb^5+^, Fe^3+^ and Ti^4+^, which derived from the EBHP after the reaction. Figure 9 shows a SEM picture of EBHP. Figure 10 shows the EDS elemental plotting of EBHP (Er, Fe, Sb, O from Er_2_FeSbO_7_ and Bi, Ti, Sb, O from BiTiSbO_6_). Figure 11 shows the EDS spectra of EBHP. It can be found from Figure 9 and Figure 10 that the larger tetragonal particulates remained with BiTiSbO_6_ and the smaller circular shaped particles belonged to Er_2_FeSbO_7_. From Figure 9 and Figure 10, the particulates of BiTiSbO_6_ were encircled by the smaller particulates of Er_2_FeSbO_7_, and these two particulates were snugly associated, which manifested the triumphant fusion of EBHP. BiTiSbO_6_ possessed rhombic dodecahedron-like morphology. As we all know, different surface energy of crystal sections gain command of the anatomical upgrowth of a photocatalyst. The professors discovered the order of the skin layer energy of dissimilar sides of silver phosphate (SP) (111) < SP (100) < SP (110). In mean solar time, the SP (110) facet showed higher skin layer energy than the (111) facet, which was the reason why it haphazardly took the form of SP architectures along (110) crystallographic orientation, leading to the formulation of a diamond dodecahedrons-like morphology for SP [59]. Thus, BiTiSbO_6_ had diamond dodecahedron-like morphology that could be interpreted as being above description. The experimentation upshots which are shown in Figure 9 signified that Er_2_FeSbO_7_ had an inerratic sphere-like morphology and a well-proportioned particulates repartition. The particulate size of Er_2_FeSbO_7_ was surveyed to be around 250 nm, while the larger particulate dimension of BiTiSbO_6_ was surveyed to be about 1800 nm.

The SEM-EDS generalization analysis effects which are shown in Figure 9, Figure 10 and Figure 11 exposited that there were no excess doped elementary substances in the EBHP. In the same breath, the pure phase of Er_2_FeSbO_7_ was accordant with the X-ray diffraction generalization analysis effect which was exhibited in Figure 1. It could be predicated from Figure 10 and Figure 11 that erbium elementary substance, ferrum elementary substance, antimony elementary substance, bismuth elementary substance, titanium elementary substance, and oxygen elementary substance were contained within EBHP. The above effects were consilient with the XPS effects of EBHP, which are shown in Figure 4 and Figure 5. These were based on the EDS spectrum of EBHP (Figure 11), on the grounds that the EDS spectrogram of EBHP, the atomistic proportion of Er:Fe:Sb:Bi:Ti:O were 492:244:834:593:611:7226, which was also accordant with the XPS effects of EBHP. The atomic rate of Er_2_FeSbO_7_:BiTiSbO_6_ was obtained as 41:100. Based on the above effects, we suggest that the EBHP were of a high purity quotient following our experimental environment and prerequisites.

### 2.4. Photocatalytic Activity

Figure 12 shows the consistence change curve of ENR during the PCD of ENR with EBH, Er_2_FeSbO_7_, BiTiSbO_6_, or N-TO as a photocatalyzer under VLGI. From Figure 12, we first adsorbed enrofloxacin in the dark with EBH, Er_2_FeSbO_7_, BiTiSbO_6_, or N-TO as a photocatalyst. It can be seen from Figure 12 that after 45 min of adsorption under dark conditions with EBH as a catalyst, the adsorption rate of ENR in drug wastewater was 3.88%. All the other experiments followed the same dark conditions for 45 min. When Er_2_FeSbO_7_ was used as a photocatalyst, the adsorption rate of ENR was 3.6%. Using BiTiSbO_6_ as a photocatalyst, the adsorption rate of ENR in pharmaceutical wastewater was 3.2%. Using N-TO as a photocatalyst, the adsorption rate of ENR was 12.12%.

As can be seen from Figure 12, when EBH, Er_2_FeSbO_7_, BiTiSbO_6_, or N-TO was used as a photocatalyst to degrade ENR, the concentration of ENR in drug waste water continually decreased with the increase in VLGI time. The effects that are attained from Figure 12 indicate that the RR of ENR within pharmaceutical waste water reached 99.16%. The velocity of the reaction was 2.75 × 10^−9^ mol/L/s and the photonic efficiency (PE) was 0.0578% with EBHP as a catalyzer after a VLGI of 150 min (VLGI-150min). All the other experiments followed the same VLGI-150min. When Er_2_FeSbO_7_ was used as a photocatalyzer, the RR of ENR reached 86.12%, the rate of the reaction was 2.39 × 10^−9^ mol/L/s and the PE was 0.0502%. The RR of ENR within pharmaceutical waste water reached 78.12%, the rate of reaction was 2.17 × 10^−9^ mol/L/s and the PE was 0.0456% with BiTiSbO_6_ as a photocatalyzer. The RR of ENR reached 39.28%, the velocity of the reaction was 1.09 × 10^−9^ mol/L/s and the PE was 0.0229% with N-TO as a photocatalyzer. The effects that are attained from Figure 12 indicate that the RR of ENR within pharmaceutical waste water reached 77.56%, and the velocity of the reaction was 3.23 × 10^−9^ mol/L/s with EBH as a photocatalyzer after VLGI of 100 min. Moreover, the results that are attained from Figure 12 represent that the RR of ENR within pharmaceutical waste water reached 99.16%, and the velocity of the reaction was 2.75 × 10^−9^ mol/L/s with EBH as a catalyst after VLGI-150min. It can be seen from the above results that the concentration of ENR decreased and the removal rate of ENR increased with prolonging the VLGI time during the PCD of ENR with EBH as the photocatalyst.

Moreover, we can summarize from the above effects that the photodegradation efficiency (PGE) of ENR in the presence of EBHP was the highest. Simultaneously, the PGE of ENR with Er_2_FeSbO_7_ as a photocatalyzer was higher than that with BiTiSbO_6_ as a photocatalyzer or with N-TO as a photocatalyzer. Ultimately, the PGE of ENR with BiTiSbO_6_ as a photocatalyzer was higher than that with N-TO as a photocatalyzer, manifesting that the VL light-catalyzed liveness of EBHP was maximally contrasted with that of Er_2_FeSbO_7_, BiTiSbO_6_ or N-TO. The above effects show that the RR of ENR with EBHP was 1.151 times, 1.269 times or 2.524 times higher than that with Er_2_FeSbO_7_, BiTiSbO_6_, or N-TO as a photocatalyzer after VLGI-150min. In the previous reports, Chen et al. found that graphene-like white graphite modified bismuth phosphate material could be used to promote the photocatalytic degradation rate of ENR. The experimental results show that the removal rate of ENR was 79.2%, 91.5% or 83.8% by 0.5 wt% BN/BiPO_4_, 1 wt% BN/BiPO_4_ or 2 wt% BN/BiPO_4_ under sunlight irradiation of 120 min. It could be found that 1wt% graphene-like BN/BiPO_4_ had the highest degradation rate of ENR [51]. However, our experimental results show that the removal rate of ENR was 95.75% with EBH as a photocatalyst under a VLGI of 120 min. Therefore, the EBHP which was studied in this paper possessed higher degradation efficiency during the degradation process of ENR compared with 1 wt% graphene-like BN/BiPO_4_.

Using Er_2_FeSbO_7_ or EBH as a photocatalyzer separately, the consistence of null valence Sb or Sb^5+^ in the water before the PCD of ENR was zero. After VLGI-150 min for the PD of ENR, the contents of null valence Sb or Sb^5+^ in the water contents was also zero. Regarding the specific surface, the surface area of Er_2_FeSbO_7_ was 4.21 m^2^/g, the surface area of BiTiSbO_6_ was 3.94 m^2^/g, and the surface area of EBHP was 4.52 m^2^/g. In addition, the average pore diameter of Er_2_FeSbO_7_, BiTiSbO_6_ or EBHP was calculated to be 13.69 nm, 14.12 nm or 13.14 nm. From Figure 12, it can be seen that the catalytic coefficient of performance of Er_2_FeSbO_7_/BiTiSbO_6_, Er_2_FeSbO_7_ and BiTiSbO_6_ after a VLGI of 100 min was 77.56%, 50.20% and 42.52%, respectively. Therefore, it could be concluded that the photocatalyst which possessed a larger specific surface area (SSA) had higher photocatalytic activity. Our PLR were expounded within several Figures by the photocatalyzer, which was arranged by the hydrothermal synthesis quomodo (HSQ), thus, it would not conduce disparity in light-catalyzed liveness. XRD mensuration was carried out to indagate the chemical composition (CC) and phase architecture of the preliminary materials. All sharp diffraction peaks of the materials which were prepared by HSQ showed that the samples were well crystallized, and an impurity peak was not observed, which indicated that the purity of the product was very high.

The size of the photocatalyst particle morphology would affect the SSA of the photocatalyst. The larger SSA of the catalyst would result in more active sites on the catalyst surface, thus, the photocatalytic activity would be stronger. When EBH was used as a catalyst, the RR of ENR could reach 99.16% after VLGI-150min. This research work was carried out to develop VL-responsive nano materials with low cost, high catalytic activity and full use of 43% VL in the solar spectrum. Ultimately, the toxic organic pollutants from pharmaceutical wastewater should be deeply purified and removed, and a safe, hygienic and pollution-free water environment should be obtained.

Figure 13 shows the consistence change curves of total organic carbon (TOC) during the PCD of ENR within pharmaceutical waste water with EBH, Er_2_FeSbO_7_, BiTiSbO_6_, or N-TO as a photocatalyzer under VLGI. The consistence of ENR continually decreased with increasing VLGI time. As can be seen in Figure 13, the RR of TOC within pharmaceutical waste water attained 94.96%, 83.94%, 74.45% or 32.83%, respectively, after VLGI-150min when EBHP, Er_2_FeSbO_7_, BiTiSbO_6_ or N-TO was used for degrading ENR. In summary, we can conclude from the above effects that the RR of TOC during the degradation of ENR in EBHP was higher than that of Er_2_FeSbO_7_, BiTiSbO_6_ or N-TO. The above results also indicate that the RR of TOC during the degradation of ENR in Er_2_FeSbO_7_ was much higher than that in BiTiSbO_6_ or N-TO, which signifies that EBHP had the maximum mineralization rate during ENR degradation compared with Er_2_FeSbO_7_, BiTiSbO_6_ or N-TO.

Figure 14 shows the concentration change curve of ENR during PCD with EBH as a photocatalyzer under VLGI for TCDT. It can be seen from Figure 14 that the RR of ENR reached 98%, 96.76% or 95.6%, respectively, after VLGI-150min with EBH as a photocatalyst, by accomplishing recursion retrogradation tests for degrading ENR. Figure 15 shows the consistency change crooked line of TOC during the PCD of ENR with EBH as a photocatalyzer under VLGI for TCDT. From Figure 15, it can be seen that the RR of TOC was 93.31%, 91.87% or 90.38%, respectively, after VLGI-150min with EBHP, by accomplishing TCDT for degrading ENR. The experimental effects which were attained from Figure 14 and Figure 15 show that the EBHP had a high anti-whip ability.

Figure 16 shows the first-order kinetics plots of PCD for ENR under VLGI with EBH, Er_2_FeSbO_7_, BiTiSbO_6_ or N-TO as a photocatalyzer. As can be seen from Figure 16, the power mechanics invariable k which was obtained from the dynamical crooked line toward ENR consistence and VLGI time with EBH, Er_2_FeSbO_7_, BiTiSbO_6_ or N-TO as a photocatalyzer reached 0.02296 min^−1^, 0.00882 min^−1^, 0.00699 min^−1^ or 0.00254 min^−1^, separately. The dynamic numeric constant k, which hails from the namic profile toward TOC consistence, reached 0.01535 min^−1^, 0.00786 min^−1^, 0.00607 min^−1^ or 0.00192 min^−1^ with EBH, Er_2_FeSbO_7_, BiTiSbO_6_ or N-TO as a photocatalyzer. The fact that the value of *K_TOC_* for deteriorating ENR was lower than the number of *K_C_* for deteriorating ENR by using the same catalyzer indicated that the photodegradation intermediates of ENR might appear in the PCD of ENR under VLGI. Meanwhile, contrasted with the other photocatalyzer, EBHP had a better mineralization workpiece ratio for ENR retrogradation.

Appendix A shows the single-order kinetics of the PCD of ENR under VLGI with EBH as a photocatalyst for three cycle degradation tests (TCDT). In the light of the results from Appendix A, the power mechanics invariable k which came from the dynamical crooked line toward ENR consistence and VLGI time with EBHP for TCDT reached 0.01786 min^−1^, 0.01536 min^−1^ or 0.01354 min^−1^. The power mechanics invariable k which derived from the dynamical crooked line toward TOC consistence and VLGI time with EBHP for TCDT reached 0.01352 min^−1^, 0.01206 min^−1^ or 0.01079 min^−1^. In the light of the experimental effects from Figure 16 and Appendix A, the PCD of ENR within pharmaceutical waste water with EBHP under VLGI conformed to the single-order reaction dynamics.

It can be seen from Appendix A that the RR of ENR lessened by 2.4% with EBH as a photocatalyzer under VLGI after TCDT, and the RR of TOC lessened by 2.93%. It did not show a significant difference for the retrogradation efficiency above TCDT, and the photocatalyzer architecture of EBHP was jarless.

Figure 17 presents the impact of diverse free radical scavengers (FRSs), for instance, benzoquinone (BQ),isopropanol (IPA) or ethylenediamine tetraacetic acid (EDTA) on the RR of ENR with EBH as a photocatalyzer under VLGI. At the start of the photocatalytic experiment, different FRSs were added to ENR resolution for determining the active species in the retrogradation process of ENR. We used isopropanol (IPA) to capture HRS (^•^OH); benzoquinone (BQ), which we utilized for captured superoxide anions (^•^O_2_^−^); and ethylenediaminetetraacetic acid (EDTA), which we used for captured holes (h^+^). The scheming IPA consistence, BQ consistence or EDTA consistence was 0.15 mmol/L, and the added amount of IPA, BQ or EDTA was 1 mL. As for the selection of free radical scavenger consistence, five concentrations of free radical scavengers (0.05 mmol/L, 0.1 mmol/L, 0.15 mmol/L, 0.2 mmol/L and 0.25 mmol/L, respectively) were used to participate in the reaction. According to the experimental results, the concentration of capture agent was taken as the abscissa, meanwhile, [99.16%—C] was taken as the ordinate. Ultimately, five corresponding curves were obtained. As for [99.16%—C], 99.16% was the removal rate of ENR at 150 min in the blank experiment, and C was the removal rate of ENR after adding the corresponding concentration of capture agent for 150 min. At this time, the curve had a maximum value, which meant that the corresponding radicals were completely captured by the capture agent. At this time, the highest point was determined as the concentration of free radical scavenger which was required in the experiment.

In light of Figure 17, while the IPA, BQ or EDTA was added into ENR solution, the RR of ENR decreased by 59.87%, 70.89% or 29.15%, respectively, compared with the RR of ENR which came from the matched troop. Thus, it could be predicated that ^•^OH, h^+^ and ^•^O_2_^−^ were all active free radicals in the process of ENR retrogradation. It could be seen from Figure 17 that ^•^OH in the ENR result played an overriding position when ENR was degraded with EBH as a photocatalyzer under VLGI. Through the medium of the project of the appending trapping agent, it was assumed that superoxide anion possessed maximal oxidation removal ability for removing ENR within pharmaceutical waste water, relative to hydroxyl radicals or holes. The oxygenation elimination capacity for degrading ENR, which was from high to low among three oxygenation radicals, was as follows: superoxide anion > hydroxyl radicals > holes.

A Nyquist impedance plot measurement was the other significant qualitative method which showed the photo-induced sigma electrons and photo-induced cavities transfer procedure of the preliminary photocatalyzer at solid/electrolyte limiting surfaces. The smaller the arc radius (ARs) was, the higher the transport workpiece ratio of the photocatalyst was. Figure 18 shows the homologous Nyquist impedance plots of the preliminary EBHP, Er_2_FeSbO_7_ or BiTiSbO_6_ photocatalyzer. It was easily found from Figure 18 that the size of the ARs was in the order: BiTiSbO_6_ > Er_2_FeSbO_7_ > EBHP. Above effects indicated that the preliminary EBHP presented a more expeditious disassociation of photo-induced sigma electrons and photo-induced holes, and a faster interfaced charge transfer capability. After VLGI of 100 min, the removal rate of ENR with EBH, Er_2_FeSbO_7_ or BiTiSbO_6_ as a catalyst was 77.56%, 50.2% or 42.52%, respectively, which was consistent with the intercomparable results of the curvature radius of the catalysts in the electrochemical impedance experiment. After a VLGI of 150 min, the removal rate of ENR by EBH, Er_2_FeSbO_7_ or BiTiSbO_6_ was 99.16%, 86.12% or 78.12%, respectively, which was also consistent with the intercomparable results of the curvature radius of the catalysts in the electrochemical impedance experiment. Therefore, it could be concluded that the smaller curvature radius of the catalyst led to a higher removal rate of ENR and a higher catalytic activity of the catalyst.

### 2.5. Possible Degradation Mechanism Analysis

Figure 19 shows the potential PCD machine process of ENR with EBH as a photocatalyst under VLGI. The potentials of VB and CB for a semi-conductor catalyzer could be calculated based on Equations (3) and (4) [59]:*E*_CB_ = *X* − *E*^e^ − 0.5*E*_g_(3)
*E*_VB_ = *E*_CB_ + *E*_g_(4)
where *E*_g_ is the BDG of semi-conductor, *X* is the electronegativity of the semi-conductor, and *E*^e^ is the energy of free electrons on the hydrogen scale (about 4.5 eV). Based on the above equations, the VB potential (VBP) or the CB potential (CBP) for Er_2_FeSbO_7_ was imputed to be 1.17 eV or −0.71 eV, separately. For BiTiSbO_6_, the VBP or the CBP was imputed to be 2.995 eV and 0.31 eV, separately. It could be discovered that both Er_2_FeSbO_7_ and BiTiSbO_6_ could absorb VL and internally generated electrons–holes pairs when the EBHP was search lighting by VL. Due to the redox potential position of the CB of Er_2_FeSbO_7_ (−0.71 eV), it was more subtractive than that of BiTiSbO_6_ (0.31 eV), and the photo-induced sigma electrons on the CB of Er_2_FeSbO_7_ could migrate to the CB of BiTiSbO_6_. the redox potential position of the VB of BiTiSbO_6_ (2.995 eV) was more nonnegative than that of Er_2_FeSbO_7_ (1.17 eV), and the photo-induced cavities on the VB of BiTiSbO_6_ could migrate to the VB of Er_2_FeSbO_7_.

Therefore, combining Er_2_FeSbO_7_ and BiTiSbO_6_ realizes a new heterojunction photocatalyst EBHP that can powerfully lessen the recombination rate of photo-induced sigma electrons and photo-induced cavities, reduce the essential resistance, extend the service life of photo-induced sigma electrons and photo-induced cavities, and improve the interface charge migration efficiency. Therefore, more oxyradicals such as ^•^OH or ^•^O_2_^−^ could be used for increasing the retrogradation workpiece ratio of ENR. Furthermore, the CBP of Er_2_FeSbO_7_ was −0.71 eV, which was more subtractive than that of O_2_/^•^O_2_^−^ (−0.33 V), indicating that the electrons within the CB of Er_2_FeSbO_7_ could assimilate O_2_ for caused ^•^O_2_^−^, which could degrade ENR (as shown as path 1 in Figure 19). Simultaneously, the VBP of BiTiSbO_6_ was 2.995 eV, which was more positive than that of OH^−^/^•^OH (2.38 V), indicating that the holes in the VB of BiTiSbO_6_ could oxidize H_2_O or OH^−^ into ^•^OH for degrading ENR, which is shown as path 2. Ultimately, the photo-induced cavities in the VB of Er_2_FeSbO_7_ or BiTiSbO_6_ could directly oxidize ENR and degrade ENR due to the high oxidizing ability, and this is shown as path 3. In conclusion, EBHP had good photocatalytic activity for the degradation of ENR, which was mainly due to the efficient electron and hole separation efficiency induced by EBHP.

With the view of studying the retrogradation mechanism of ENR, the in-between products yield which were caused during the retrogradation process of ENR were checked by LC–MS. The midterm offspring which were attained during the PCD of ENR were identified as C_17_H_16_FN_3_O_5_ (*m/z* = 362); C_16_H_19_N_3_O_3_ (*m/z* = 302); C_15_H_18_N_2_O_3_ (*m/z* = 275); C_13_H_10_N_2_O_3_ (*m/z* = 242); C_10_H_6_N_2_O_4_ (*m/z* = 218); C_10_H_5_N_2_O_3_ (*m*/*z* = 202); C_5_H_12_NO_2_ (*m*/*z* = 118); 2-hydroxypropanoic acid; 2-hydroxyacetic acid; and 3-hydroxypropanoic acid. According to the above checked in-between products yield, the PCD paths of ENR were proposed. Figure 20 shows the PCD pathway scheme recommended by ENR under VLGI with EBH as a photocatalyzer. It can be discovered from Figure 20 that an oxidizing reaction and hydroxylation reaction were sensed during the PCD process of ENR. Finally, ENR was transformed into a micromolecule organic compound, and combined with oxidative active free radicals to transform into CO_2_ and H_2_O.

## 3. Experimental Section

### 3.1. Materials and Reagents

Ethylene diamine tetraacetic acid (EDTA, C_10_H_16_N_2_O_8_, purity = 99.5%) and isopropanol (IPA, C_3_H_8_O, purity ≥ 99.7%) were used at an analytic degree. Benzopuinone (BQ, C_6_H_4_O_2_, purity ≥ 98.0%) was used at a chemical degree. The above reagent chemicals were purchased from Sinopharm Group Chemical Reagent Co., Ltd. (Shanghai, China). Straight alcohol (C_2_H_5_OH, purity ≥ 99.5%), in keeping with American Chemical Society Specifications, was purchased from Aladdin Group Chemical Reagent Co., Ltd. (Shanghai, China). ENR (C_19_H_22_FN_3_O_3_, purity ≥ 98%) was used at a gas-phase chromatography degree as the model material, and purchased from Tianjin Bodi Chemical Co., Ltd., Tianjin, China. Ultrapure water (UPW) (18.25 MU cm) was used from throughout this work.

### 3.2. Preparation Method of Er_2_FeSbO_7_

A novel photocatalyzer, Er_2_FeSbO_7_, was compounded by the high temperature solid-state sintering method. Er_2_O_3_, Fe_2_O_3_ and Sb_2_O_5_ (Sinopharm Group Chemical Reagent Co., Ltd., Shanghai, China) with a purity of 99.99% were used as rough stock without further purification. All powders (n(Er_2_O_3_):n(Fe_2_O_3_):n(Sb_2_O_5_) = 2:1:1) were synthesized after drying at 200 °C for 4 h. In order to prepare Er_2_FeSbO_7_, the precursor was stoichiometric blended, then pushed into a small column and fitted into a compalox crucible (Shenyang Crucible Co., LTD, Shenyang, China). After scorifying at 400 °C for 2 h, the primary materials and the small pillars were taken out of the galvanical stove. The compounded materials were ground and then fitted into the galvanical furnace (KSL 1700X, Hefei Kejing Materials Technology CO., LTD, Hefei, China). Finally, calcination was carried out in an electric smelter at 1100 °C for 36 h.

A total of 0.30 mol/L of Er(NO₃)₃·5H₂O, 0.15 mol/L of Fe(NO_3_)_3_ and 0.15 mol/L of SbCl_5_ were compounded and stirred for 20 h. This was transferred to an autoclave lined with polyfluortetraethylene and heated at 200 °C for 15 h. Subsequently, the attained powder was scorified at 800 °C for 10 h in a tube furnace at a velocity of 8 °C/min under an atmosphere of N_2_. Finally, Er_2_FeSbO_7_ powder was also attained by HSQ.

### 3.3. Preparation Method of BiTiSbO_6_

A novel photocatalyzer, BiTiSbO_6_, was compounded by the high temperature solid-state sintering method. Bi_2_O_3_, TiO_2_ and Sb_2_O_5_ (Sinopharm Group Chemical Reagent Co., Ltd., Shanghai, China) with a purity of 99.99% were used as rough stock without further purification. All powders (n(Bi_2_O_3_):n( TiO_2_):n(Sb_2_O_5_) = 1:2:1) were synthesized after drying at 200 °C for 2 h. In order to prepare BiTiSbO_6_, the precursor was stoichiometric blended, then pushed into a small column and fitted into an compalox crucible (Shenyang Crucible Co., Ltd., Shenyang, China). After scorifying at 400 °C for 4 h, the raw material and small cylinder were removed from the galvanical furnace. The compounded materials were ground and then fitted into the galvanical furnace (KSL 1700X, Hefei Kejing Materials Technology CO., Ltd., Hefei, China). Finally, calcination was carried out in an electric smelter at 1010 °C for 25 h.

A total of 0.15 mol/L of Bi(NO_3_)_3_·5H_2_O, 0.15 mol/L of TiCl_4_, and 0.15 mol/L of SbCl_5_ was compounded and stirred for 20 h. The solution was transferred into a Teflon-lined autoclave and scorified at 200 °C for 15 h. Afterwards, the achieved powder was scorified at 780 °C for 10 h in a tube-type furnace at a velocity of 8 °C/min under an air envelope of N_2_. BiTiSbO_6_ powder was finally achieved.

### 3.4. Synthesis of N-Doping TiO_2_

Nitrogen-doped titania (N-doping TiO_2_) catalyst was prepared by sol-gel modus with tetrabutyl titanate as a precursor and ethanol as a solvent. The procedure was as follows: first, 17 mL of tetrabutyl titanate and 40 mL of absolute ethyl alcohol were combined to serve as solution A; 40 mL of absolute ethyl alcohol, 10 mL of glacial acetic acid and 5 mL of double distilled water were blended to be solution B; subsequently, solution A was added dropwise into solution under vigorous magnetic stirring condition, and then a transparent colloidal suspension was shaped. Then, ammonial solution with an N/Ti ratio of 8 mol% was fitted into the transparent tremellose soliquid under magnetical mix round prerequisite for 1 h. Dry gel was then generated after 2 days of aging. The xerogels were ground into powder, dead-burned at 500 degrees for 2 h, then ground on the powder and screened by a vibrating screen to obtain N-TO powder.

### 3.5. Synthesis of Er_2_FeSbO_7_/BiTiSbO_6_ Heterojunction Photocatalyst

The maximal calcination temperature (MCT) of Er_2_FeSbO_7_ which was prepared by solid state sintering modus was 1100 °C, and the insulated time was 36 h. The MCT of BiTiSbO_6_ which was preliminary by solid state sintering modus was 1010 °C, and the insulated time was 25 h. The MCT of BiTiSbO_6_ which was preliminary by HSQ was 780 °C, and the insulated time was 10 h. The MCT of Er_2_FeSbO_7_ which was preliminary by HSQ was 800 °C, and the insulated time was 10 h. The higher the MCT, the greater the power dissipation energy, which would decrease and expend the operational life proof cycle of the stove. The longer insulated time and the higher maximal sintering temperature would cause the larger particle size of BiTiSbO_6_ or Er_2_FeSbO_7_. That being so, the SSA of BiTiSbO_6_ or Er_2_FeSbO_7_ would be reduced and the light-catalyzed liveness of BiTiSbO_6_ or Er_2_FeSbO_7_ would be accordingly lessened. For the purpose of increasing the light-catalyzed liveness, reducing energy consumption and increasing the instrument operational life proof cycle of high-temperature incinerator, we used HSQ for preparing BiTiSbO_6_ and Er_2_FeSbO_7_ in the process of preparing HJ.

Er_2_FeSbO_7_ and BiTiSbO_6_ were prepared by HSQ, which mainly used the dissolution recrystallization mechanism to dissolve Er (NO_3_)_3_·5H_2_O; Fe(NO_3_)_3_; SbCl_5_; Bi(NO_3_)_3_·5H_2_O; and TiCl_4_ in hydrothermal medium, and then the above materials entered the solution in the form of ion groups and molecular groups. Strong convection, which was caused by a temperature difference in autoclave, would prompt these ions and molecules to transport to the growth area, which contained seed crystal. Ultimately, the saturated solution was formed and crystallized.

First, 0.30 mol/L of Er (NO_3_)_3_·5H_2_O; 0.15 mol/L of Fe(NO_3_)_3_ and 0.15 mol/L of SbCl_5_ were compounded and stirred for 20 h. This solution was migrated into a polyfluortetraethylene sterilizer and scorified at 200 °C for 15 h. Afterward, the achieved powder was scorified at 800 °C for 10 h in a tubular furnace at a velocity of 8°C/min under an atmosphere of N_2_. Er_2_FeSbO_7_ powder was finally attained. Secondly, 0.15 mol/L of Bi(NO_3_)_3_·5H_2_O; 0.15 mol/L of TiCl_4_; and 0.15 mol/L of SbCl_5_ were mixed and stirred for 20 h. This solution was migrated into a polyfluortetraethylene sterilizer and heated at 200 °C for 15 h. Afterward, the achieved powder was calcined at 780 °C for 10 h in a tubular furnace at a velocity of 8 °C/min under an air envelope of N_2_. BiTiSbO_6_ powder was finally attained.

The prepared Er_2_FeSbO_7_ and BiTiSbO_6_ powders were prepared by the solvothermal method. The powders of Er_2_FeSbO_7_ or BiTiSbO_6_ were prepared by dissolving in octanol organic solvent in an autoclave. At this time, under liquid phase and supercritical conditions, the reactants were dispersed in the solution and became more active. Therefore, the reactants were dissolved and dispersed, meanwhile, the reaction occurred, and the product was synthesized slowly.

A handy solvothermal modus was used for synthesizing new EBHP in this report. EBHP was preliminary by compounding 525.36 mg of Er_2_FeSbO_7_ with 30 wt% (974.64 mg) of BiTiSbO_6_ in 300 mL of octanol (C_8_H_18_O) and then dispersed in an ultrasonic bath for 1 h. Then, it was tepefied and channeled back at 140 °C for 2 h under acute whisking condition to increase the adhesion of Er_2_FeSbO_7_ on the surface of BiTiSbO_6_ nanoparticles to form EBHP. After cooling to indoor temperature, the product was attained by the centrifuge method and rinsed several times with a mixture of n-hexane/ethanol. The purified powder was dried in a vacuum oven at 60 °C for 6 h and stored in a dryer for further use. Finally, EBHP was successfully prepared.

### 3.6. Characterizations

The pure crystals of the preliminary patterns were checked by the powder X-ray diffractometer (XRD, Shimadzu, XRD-6000, Cu Kα radiation, λ = 1.54184 Å, sampling pitch of 0.02°, preset time of 0.3 s step^−1^, Kyoto, Japan). The morphology and microstructure of the preliminary patterns were characterized by using a scanning electronic microscope (SEM, FEI, Quanta 250), and the elementary composition which derived from above prepared samples was captured by energy dispersive spectroscopy (EDS). The diffuse reflectance spectrum of the above prepared sample was obtained by using an UV-Vis spectrophotometer (UV-Vis DRS, Shimadzu, UV-3600, Kyoto, Japan). Surface CC and states of the prepared sample were analyzed by X-ray photoelectron spectrograph (XPS, UlVAC-PHI, PHI 5000 VersaProbe) with an Al-kα X-ray source.

### 3.7. Photoelectrochemical Experiments

Electrochemical impedance spectroscopy (EIS) project was implemented by a CHI660D electrochemical station (Chenhua Instruments Co., Shanghai, China) with a normative 3-electrode. The 3-electrode comprised a working electrode (as-prepared catalysts); counter electrode (platinum plate); and reference electrode (Ag/AgCl electrode). The electrolyte was Na_2_SO_4_ aqueous solution (0.5 mol/L). The lamphouse for the project was a 500 W Xe lamp with an UV cut-off filter. We placed the pattern (0.03 g) and chitosan (0.01 g) in dimethyl formamide (0.45 mL) and then used ultrasonication for an hour to obtain the well-proportioned suspending liquid. Afterward, they were trickled on indium tin oxide (ITO) conducting glass (10 mm × 20 mm). Finally, the working electrode was dried at 80 °C for 10 min.

### 3.8. Experimental Setup and Procedure

The projects were implemented in a photocatalytic catalyst case (XPA-7, Xujiang Electromechanical Plant, Nanjing, China) and the temperature of the response setup was 20 °C, which was mastered by rotative cooling water. Imitated solar light search-lighting was offered by a 500 W xenon with a 420 nm cut-off filter. There were 12 of the same silica tube, among which the bulk of a single response solution was 40 mL, and the total response bulk for pharmaceutic waste water was 480 mL. The doses of Er_2_FeSbO_7_, BiTiSbO_7_ or EBHP were 0.75 g/L. Furthermore, the consistence of ENR was 0.025 mmol/L. The consistence of ENR was the remaining consistence after biologic degradation for practical pharmaceutical waste water which contained an ENR of 1.0 mmol/L (Taihu Lake, Wuxi, China). During the response, 3 mL suspension was withdrawn periodically, whereafter the percolation (0.22 μm PES polyethersulfone filter membrane) was realized for removing the activator. Finally, the remaining consistency of ENR in the solution was determined by the UV-visible spectrophotometer (UV-2550, Shimadzu Corporation, Kyoto, Japan). The absorption wavelength (detecting wavelength) of ENR was 276 nm. The absorbancy normative curvilineal of ENR at dissimilar consistency was accomplished under UV-irradiation search-lighting in the area of 220–320 nm with a visible UV-irradiation wedge photometer. The relation between the consistency of ENR and the absorbancy number at 276 nm should be counted. The absorbancy of ENR in the solution was surveyed at the absorptive wavelength of 276 nm, and the specification curve of ENR was drawn and a linear regression quomodo was used for the quantification of ENR. Prior to VLGI, the suspending, which contained the photocatalyzer and ENR, was magnetically mixed in darkness for 45 min to ensure the establishment of an adsorption/desorption equilibrium among the photocatalyzer, ENR and atmospheric oxygen. During VL illumination, the suspending liquid was stirred at 500 rpm.

The mineralization project data of ENR within the response solution were surveyed by using a TOC analyzer (TOC-5000 A, shimadzu Corporation, Kyoto, Japan). For the purpose of examining the consistency of TOC during the PCD of ENR, potassium acid phthalate (KHC_8_H_4_O_4_) or natrium carbonicum calcinatum was used as normative reactant. Normative solutions of KHC_8_H_4_O_4_ with a known carbon consistency (in the range of 0–100 mg/L) were preliminary for calibration. Six samples which contained 45 mL of response solution were used for measuring TOC consistence every time.

The discrimination and survey of ENR and its in-between retrogradation offspring were carried out by liquid chromatography–mass spectrometry (LC-MS, Thermo Quest LCQ Duo, Thermo Fisher Scientific Corporation, Bay State Waltham, MA, USA. Beta Basic-C18 HPLC column: 150 × 2.1 mm, ID of 5 μm, Thermo Fisher Scientific Corporation, MA, USA). Here, 20 μL of solution attained after the photocatalytic response was emptied voluntarily into the LC–MS system. The mobile phase contained 60% methyl alcohol and 40% UPW, and the current velocity was 0.2 mL/min. MS conditions contained an electrojet electric dissociation limiting surface, a capillary temperature of 27 °C with an electric tension of 19.00 V, a spray electric tension of 5000 V and an invariable sheath gas flow velocity. The action spectrum was attained in the anion scan mode and the *m*/*z* sweep interval from 50 to 600.

For the purpose of survey, the photon intensity of incoming ray, the strainer which was 7 cm in length and 5 cm in breadth was chosen to be irradiated by incident single pitch of waves VL of 420 nm. In light of the equation of *υ* = *c*/*λ* and hv which deputized the photon energy, Avogadro’s number *N_A_*, Planck constant *h*, photonic frequency *υ*, incoming ray pitch of waves *λ* and speed of light *c* were used to attain the mole number of the total photons or the reactive photons which passed through the gross area of above strainer per unit time. To adopt adjusting the length between the photoreactor and the xenon arc lamp, the incident photon gamma flux on the photoreactor was varied.

The incident photon gamma flux Io surveyed by a radiometer-type receiver (Model FZ-A, Photoelectric Instrument Factory Beijing Normal University, Beijing, China) was determined to be 4.76 × 10^−6^ Einstein L^−1^ s^−1^ under VLGI (pitch of waves area of 400–700 nm). The incident photon gamma flux on the photoreactor was varied by adjusting the length between the photoreactor and the Xe arc lamp.

The PE was obtained based on the following Equation (5):*ϕ* = *R/I_o_*
(5)
where *ϕ* was the PE (%), *R* was the retrogradation velocity of ENR (mol L^−1^ s^−1^), and *I_o_* was the incident photon gamma flux (Einstein L^−1^ s^−1^).

## 4. Conclusions

Er_2_FeSbO_7_ was prepared by solid-state modus. For the first time, EBHP was prepared by a facile solvothermal modus. The optical physics characteristics of the single phase Er_2_FeSbO_7_ and EBH were researched and verified with SEM, XRD, UV-Vis DRS and XPS tests. The main conclusion is that Er_2_FeSbO_7_ is a pure phase which crystallizes in a pyrite architecture that maintains a cubic crystal system with the space group *Fd3m*. The crystal parameter or the BDG of Er_2_FeSbO_7_ were a = 10.179902 Å or 1.88 eV. EBHP was circumstantiated to be an efficient photocatalyzer for exterminating ENR in the waste water. After VLGI-150min, the RR of ENR or TOC reached 99.16% or 94.96%. The RR of ENR with EBH as a photocatalyzer was 1.151 times, 1.269 times or 2.524 times higher than that with Er_2_FeSbO_7_, BiTiSbO_7_ or N-TO as a catalyzer. The photocatalytic activity, which ranged from high to low among the above four photocatalysts, was as follows: EBHP > Er_2_FeSbO_7_ > BiTiSbO_6_ > N-TO. It could be proved that the oxidation removal ability for degrading ENR, which ranged from high to low among three oxidation radicals groups, was as follows: superoxide anion > HRS > holes. Therefore, it can be concluded that using EBH as a photocatalyst might be a potent method for treating pharmaceutical wastewater that is polluted by ENR. Finally, the possible photodegradation pathway for ENR was speculated.

## Figures and Tables

**Figure 1 materials-15-05906-f001:**
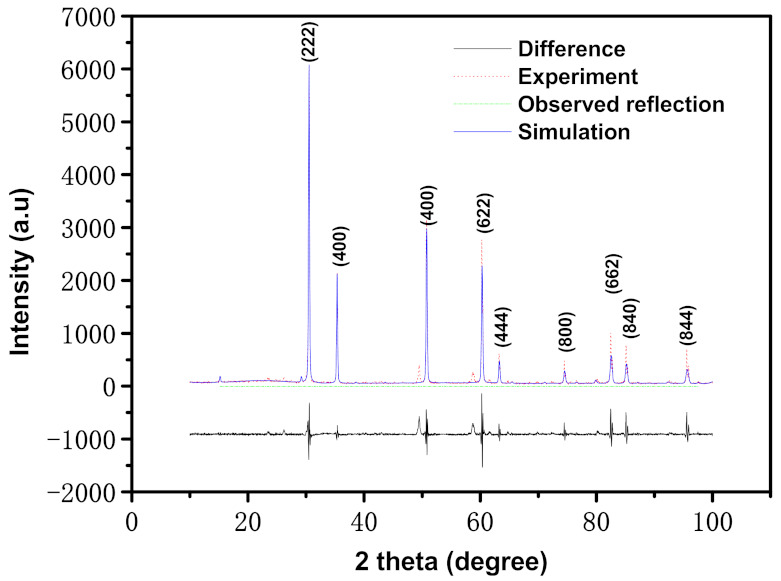
X-ray powder diffraction (XRD) patterns and Rietveld refinements of Er_2_FeSbO_7_ (red dotted line represents experimental XRD data of Er_2_FeSbO_7_; blue solid line represents simulative XRD data of Er_2_FeSbO_7_; black solid line represents a difference between experimental XRD data of Er_2_FeSbO_7_ and simulative XRD data of Er_2_FeSbO_7_; green vertical line represents observed reflection positions).

**Figure 2 materials-15-05906-f002:**
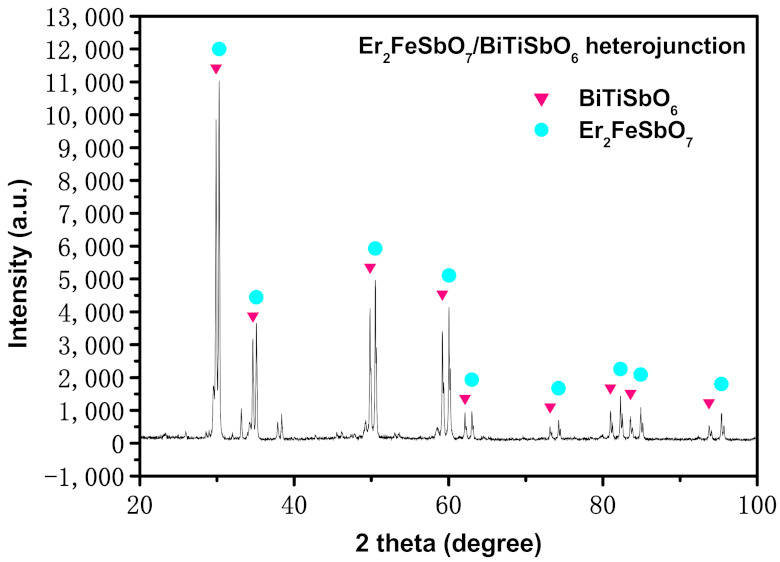
The X-ray diffraction spectrum of Er_2_FeSbO_7_/BiTiSbO_6_ HJ before reaction.

**Figure 3 materials-15-05906-f003:**
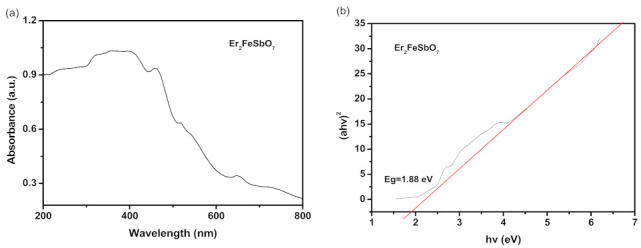
(**a**) UV-Vis diffuse reflectance spectra of Er_2_FeSbO_7_; (**b**) Plot of (*α**hν*) *^2^* versus *hν* for Er_2_FeSbO_7_.

**Figure 4 materials-15-05906-f004:**
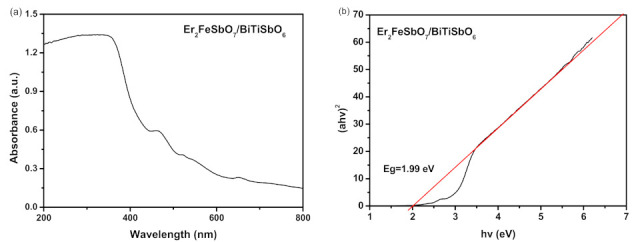
(**a**) UV-Vis diffuse reflectance spectra of Er_2_FeSbO_7_/BiTiSbO_6_ heterojunction; (**b**) plot of (*α**hν*)^1/2^ versus *hν* for Er_2_FeSbO_7_/BiTiSbO_6_ heterojunction.

**Figure 5 materials-15-05906-f005:**
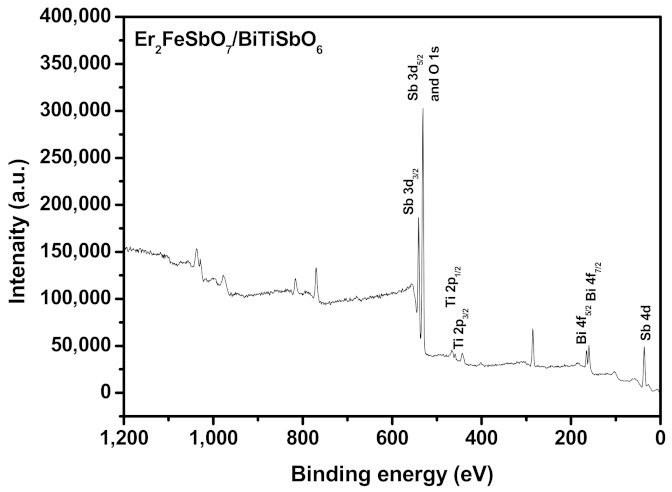
XPS survey spectrum of the EBHP before the reaction.

**Figure 6 materials-15-05906-f006:**
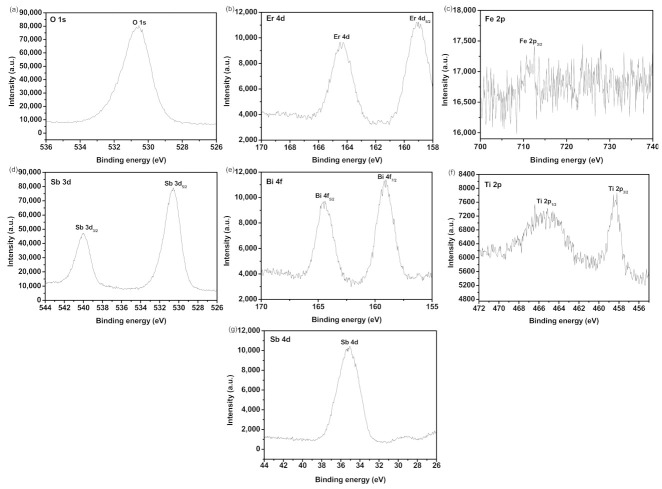
(**a**) XPS spectra of O^2−^ which derives from the EBHP before the reaction; (**b**) XPS spectra of Er^3+^ which derived from the EBHP before the reaction; (**c**) XPS spectra of Fe^3+^ which derived from the EBHP before the reaction; (**d**) XPS spectra of Sb^5^^+^ (Sb 3d) which derived from the EBHP before the reaction; (**e**) XPS spectra of Bi^3+^ which derived from the EBHP before the reaction; (**f**) XPS spectra of Ti^4+^ which derived from the EBHP before the reaction; (**g**) XPS spectra of Sb^5+^ (Sb 4d) which derived from the EBHP before the reaction.

**Figure 7 materials-15-05906-f007:**
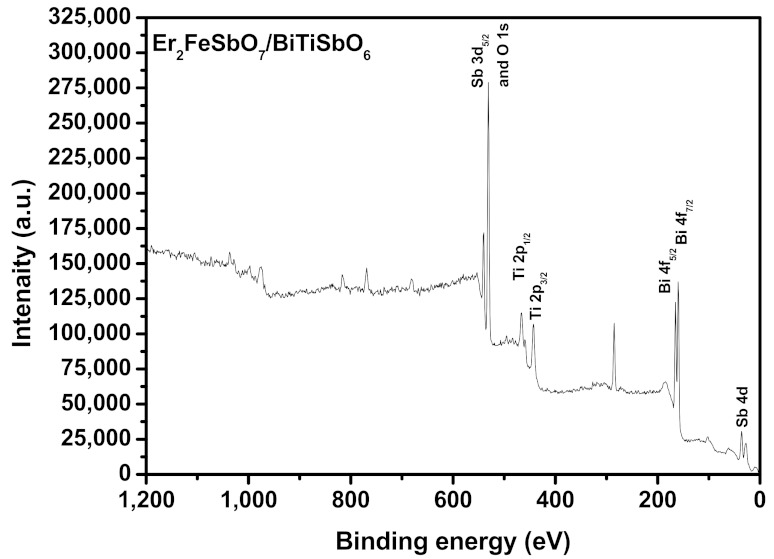
The XPS survey spectrum of the EBHP after the reaction.

**Figure 8 materials-15-05906-f008:**
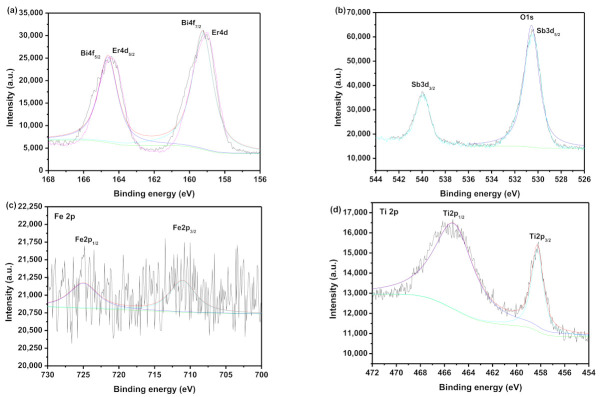
(**a**) XPS spectra of Er^3+^ and Bi^3+^ which derived from the EBHP after the reaction; (**b**) XPS spectra of O^2−^ and Sb^5+^ which derived from the EBHP after the reaction; (**c**) XPS spectra of Fe^3+^ which derived from the EBHP after the reaction; (**d**) XPS spectra of Ti^4+^ which derived from the EBHP after the reaction.

**Figure 9 materials-15-05906-f009:**
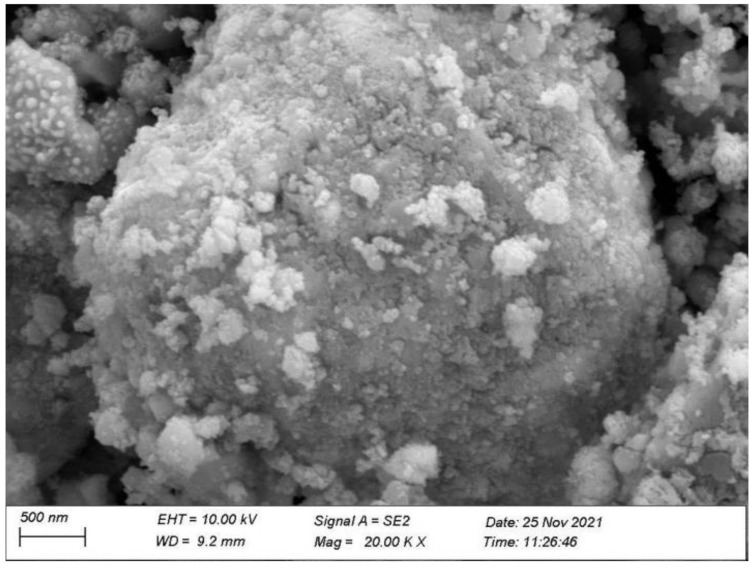
SEM photograph of EBHP.

**Figure 10 materials-15-05906-f010:**
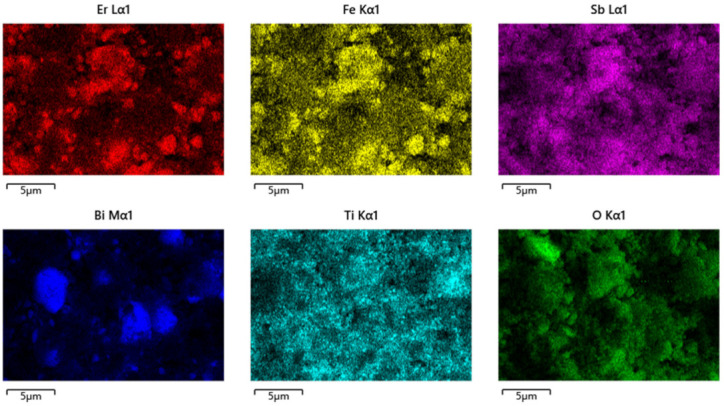
EDS elemental mapping of EBHP (Er, Fe, Sb, O from Er_2_FeSbO_7_ and Bi, Ti, Sb, O from BiTiSbO_6_).

**Figure 11 materials-15-05906-f011:**
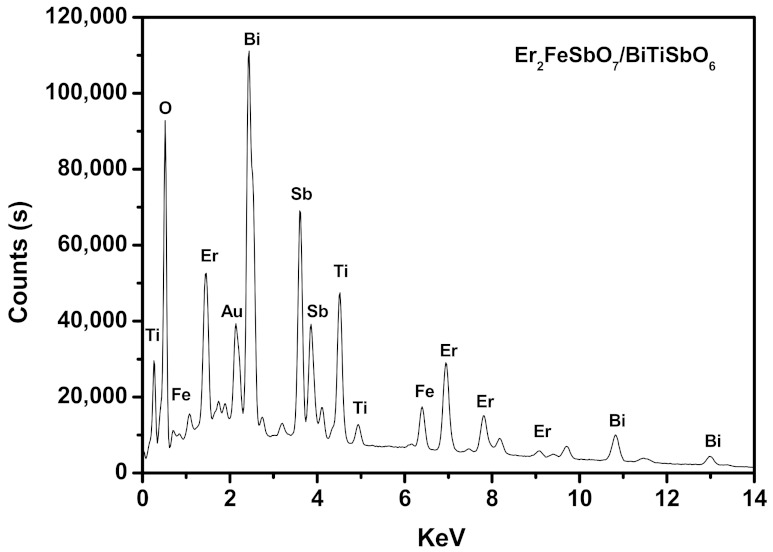
EDS spectrum of EBHP.

**Figure 12 materials-15-05906-f012:**
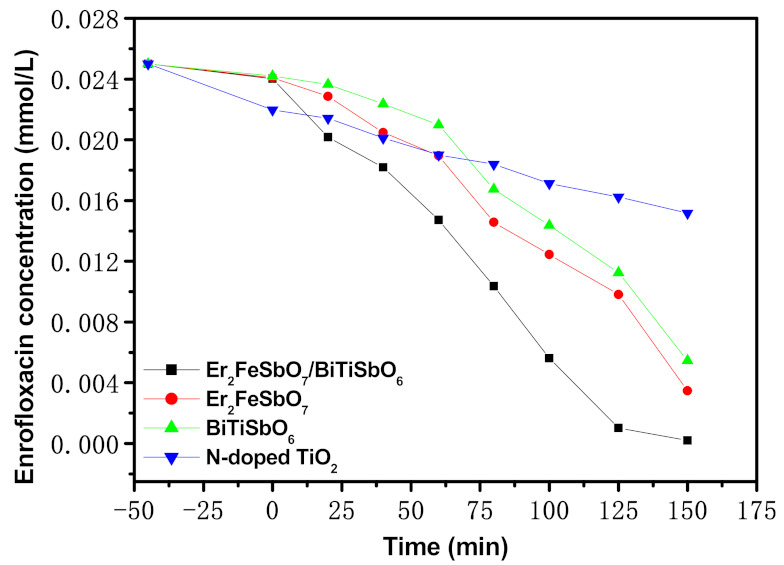
Concentration change curves of ENR during PCD of ENR with EBH, Er_2_FeSbO_7_, BiTiSbO_6_, or N-TO as photocatalyzer under VLGI.

**Figure 13 materials-15-05906-f013:**
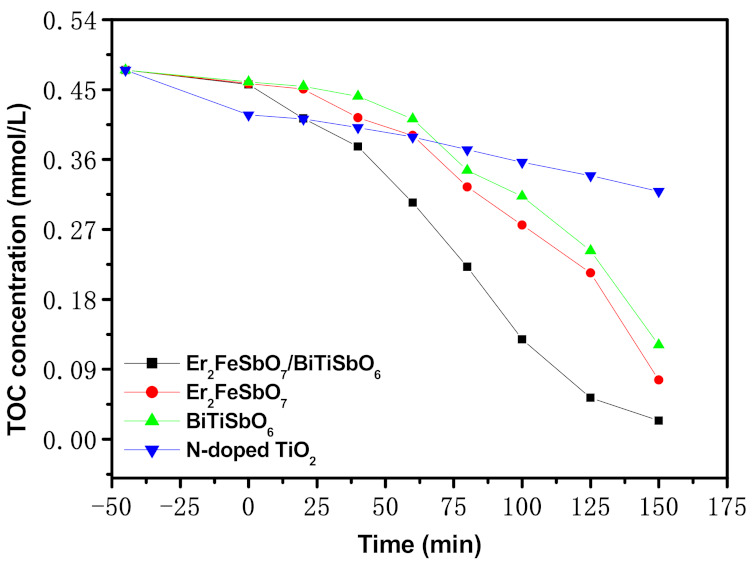
Concentration change curves of TOC during PCD of ENR in pharmaceutical waste water with EBH, Er_2_FeSbO_7_, BiTiSbO_6_ or N-TO as photocatalyzer under VLGI.

**Figure 14 materials-15-05906-f014:**
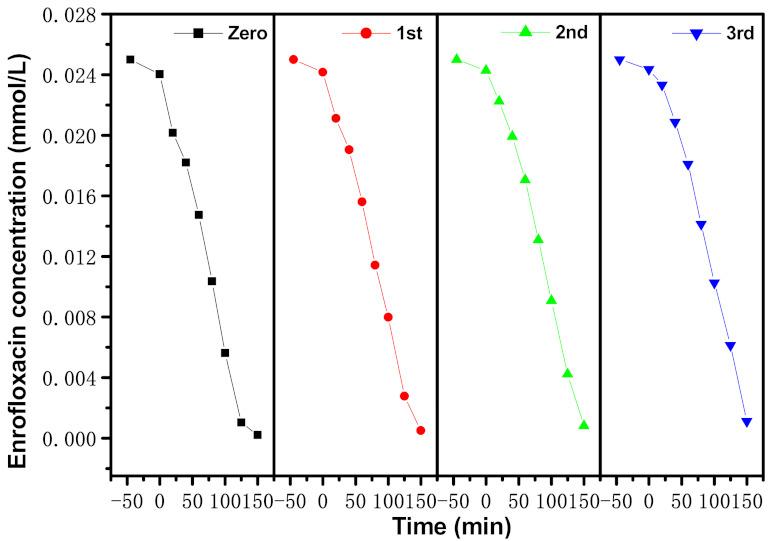
Concentration change curves of ENR during PCD of ENR within pharmaceutical waste water with EBH as photocatalyzer under VLGI for three recursion retrogradation tests.

**Figure 15 materials-15-05906-f015:**
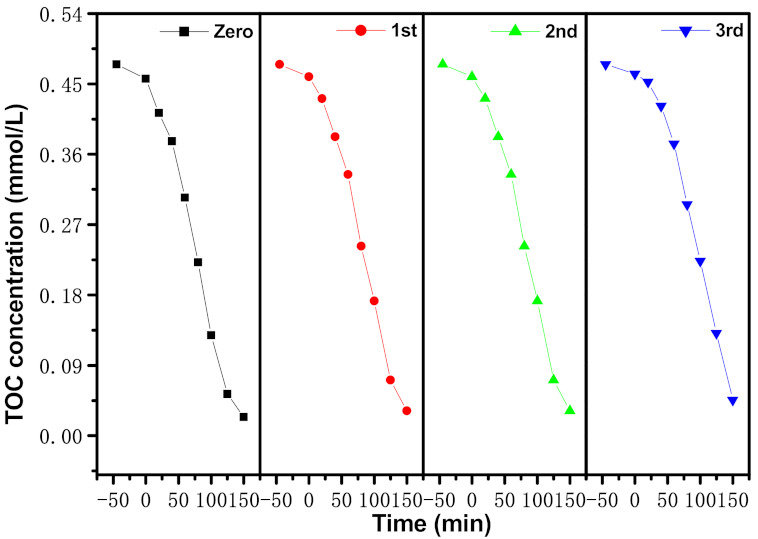
Consistence change curves of TOC during PCD of ENR within pharmaceutical waste water with EBH as photocatalyzer under VLGI for three recursion retrogradation tests.

**Figure 16 materials-15-05906-f016:**
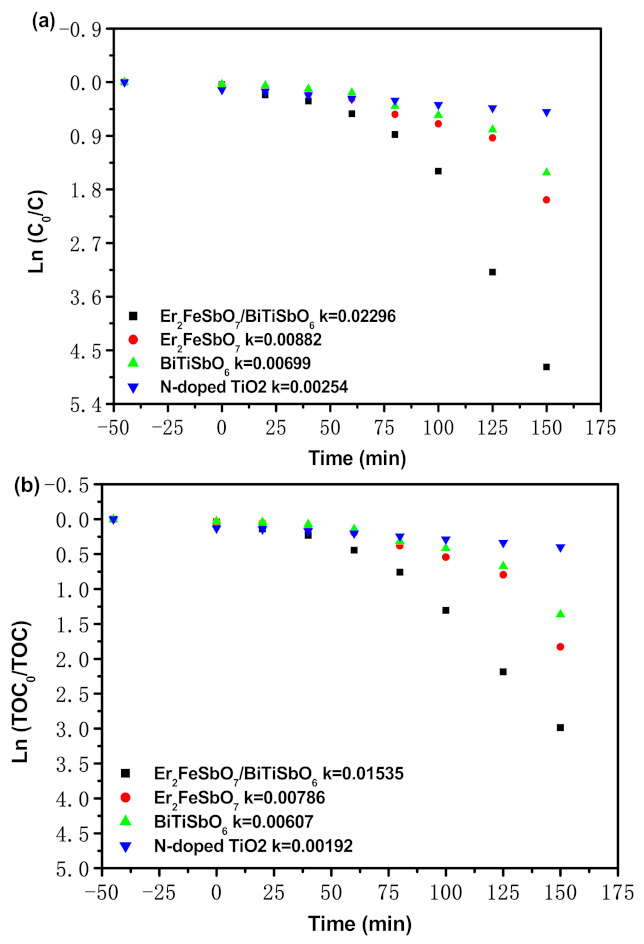
(**a**) Observed single-order kinetic plots for the PCD of ENR with EBH, Er_2_FeSbO_7_, BiTiSbO_6_ or N-TO as photocatalyzer under VLGI. (**b**) Observed single-order kinetic plots for TOC during PCD of ENR in pharmaceutical waste water with EBH, Er_2_FeSbO_7_, BiTiSbO_6_ or N-TO as photocatalyzer under VLGI.

**Figure 17 materials-15-05906-f017:**
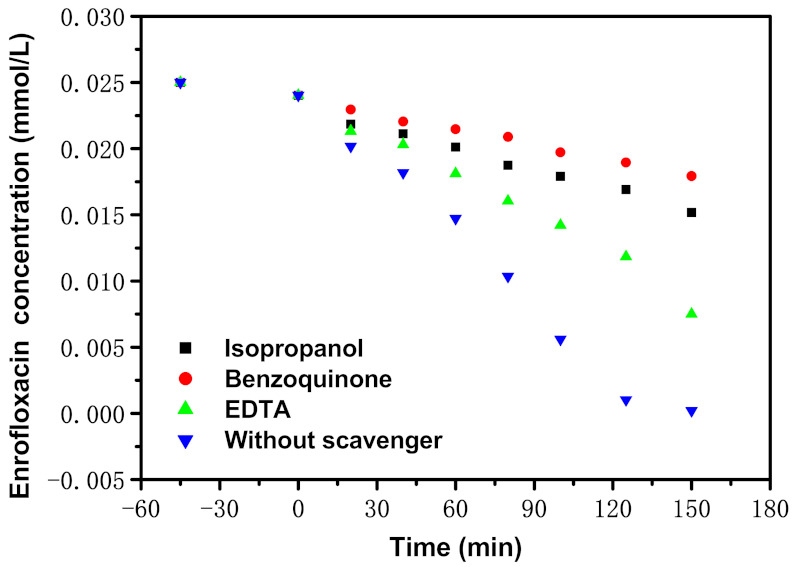
Effect of different radical scavengers such as benzoquinone (BQ), isopropanol (IPA) or ethylenediamine tetraacetic acid (EDTA) on removal efficiency of ENR with EBH as photocatalyst under VLGI.

**Figure 18 materials-15-05906-f018:**
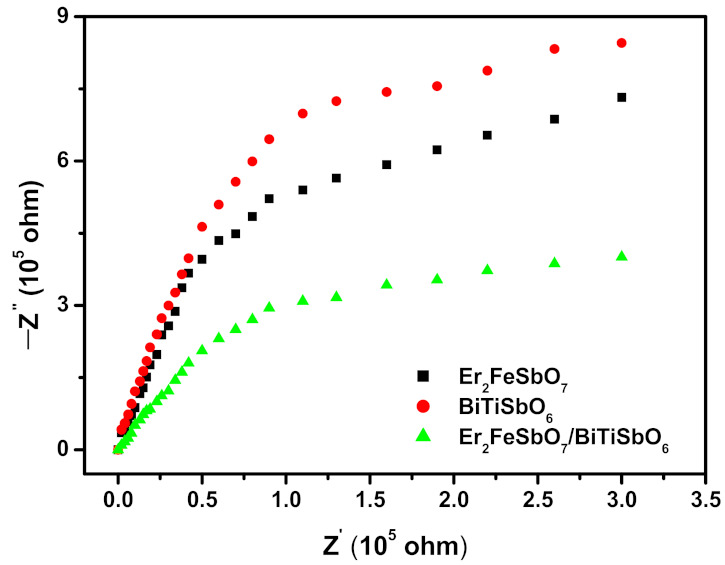
Nyquist impedance plots of EBHP, Er_2_FeSbO_7_ or BiTiSbO_6_ photocatalyst.

**Figure 19 materials-15-05906-f019:**
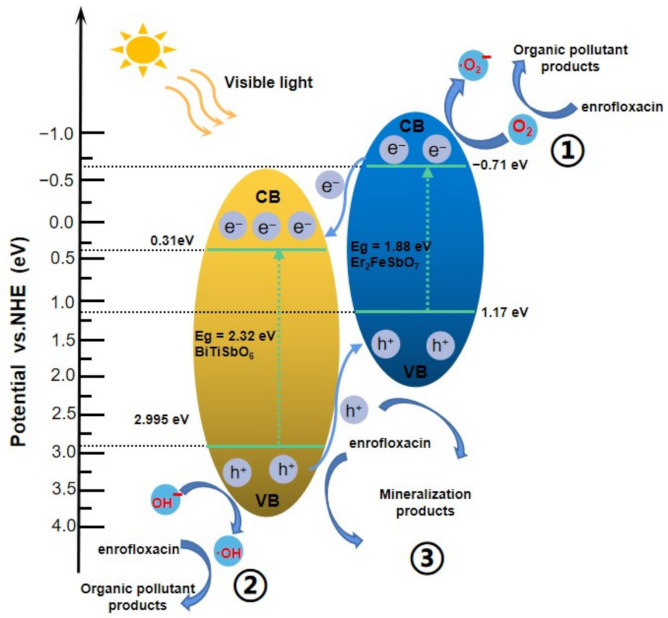
Possible PCD mechanism of ENR with EBH as photocatalyzer under VLGI.

**Figure 20 materials-15-05906-f020:**
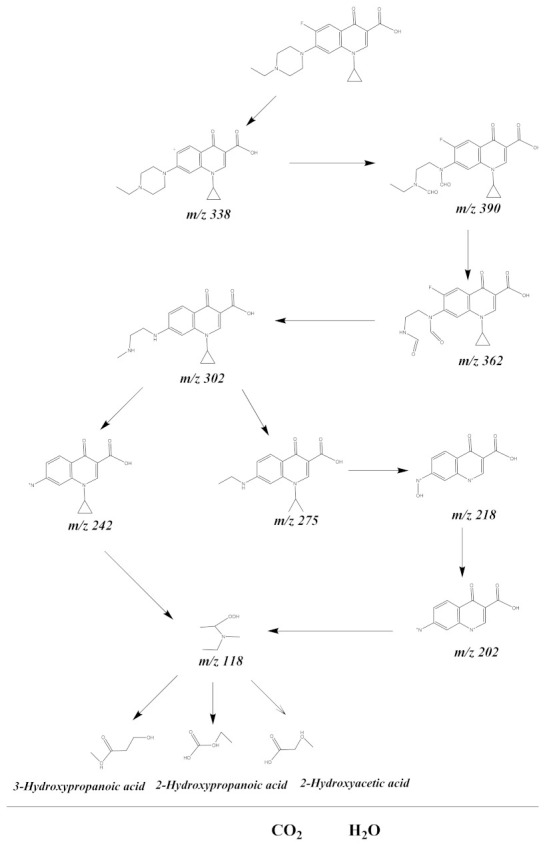
Suggested PCD pathway scheme for ENR under VLGI with EBH as photocatalyst.

**Table 1 materials-15-05906-t001:** Structural parameters of Er_2_FeSbO_7_ prepared by solid reaction process.

Atom	x	y	z	Occupation Factor
Er	0	0	0	1
Fe	0.5	0.5	0.5	0.5
Sb	0.5	0.5	0.5	0.5
O(1)	−0.185	0.125	0.125	1
O(2)	0.125	0.125	0.125	1

## Data Availability

Not Applicable.

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
