# Peer review of "Synthesis and Property Examination of Er2FeSbO7/BiTiSbO6 Heterojunction Composite Catalyst and Light-Catalyzed Retrogradation of Enrofloxacin in Pharmaceutical Waste Water under Visible Light Irradiation"

_materials, 2022, doi:10.3390/ma15175906_

Round 1
Reviewer 1 Report
Reviewer’s comments
Manuscript Number: materials-1560688
Title: Synthesis, property examination of Er2FeSbO7/BiTiSbO6 heterojunction composite catalyst and photocatalytic degradation of enrofloxacin within pharmaceutical wastewater under visible light irradiation
Journal: Materials
The authors present a manuscript focusing on the preparation and characterization of Er2FeSbO7/BiTiSbO6 heterojunction composite catalyst and their photocatalytic activity. Here are some comments that need to be addressed:
- The abstract is too long (562 words), it should highlighet only main findings, thus it must be reduced to around 200 words.
- The UV specta and energy gap calculations should be provided also for BiTiSbO6 and the composite materials not only Er2FeSbO7 (Fig. 3).
- The XPS data (Fig. 5) should be fitted to get the accurate binding energy values.
- It is recommended that the reusability and stability of the photocatalyst should be studied.
- The correlation between structural and morphological findings and photocatalytic performance should be discussed.
- The findings should be compared with pure samples to determine the composite counterparts' contribution.
Based on the comments given above, major revision is needed.
Reviewer 2 Report
Comments to author
- Abstract must be enriched via valuable results which pave the way for understanding the audiences. The current abstract is too long.
-Define all acronyms on first use in the abstract and again in the body of the manuscript.
-Fig. 5, in XPS, figures are not clear at al. Author should additionally study the deconvolution for each spectrum.
-Fig. 8, there are some unidentified peaks (such as 2.2eV, 3.8eV, etc) appeared in the spectrum of EDS. Author should identify and clearly tell about these peaks.
-The introduction is ok. Some refinement is needed with the recent work done on photocatalysis by Er2FeSbO7/BiTiSbO6 EBHP for ENR.
-Typos and grammatical error should be checked in the text.
-What the wave length for visible light used in the Photocatalytic Activity?
- Results must be discussed with the related literature.
-It is best to give the XPS test results before and after the catalytic reaction with Er2FeSbO7/BiTiSbO6 EBHP on ENR.
-For better comparison, the specific catalytic efficiency of Er2FeSbO7/BiTiSbO6 EBHP on ENR degradation should be calculated and given.
-The mechanism of enhanced photo-catalytic properties of Er2FeSbO7/BiTiSbO6 EBHP requires detailed and in-depth experimental comparison and discussion by comparing with other nanomaterials.
-How do you control growth of Er2FeSbO7/BiTiSbO6 EBHP? Give and explain the growth mechanism of Er2FeSbO7/BiTiSbO6 EBHP in text.
-What is the effect of irradiation time of ENR degradation? What is the rate of reaction?
-Synthesis of Er2FeSbO7/BiTiSbO6 EBHP material by proposed technique should be clearly and more explained.
-Result and discussion section must explain with more experimental findings, such as experimental results, significance of work, finding of results, choice of Er2FeSbO7/BiTiSbO6 EBHP materials and their practical validations.
-The figure captions are incomplete. It should be possible to understand the figure without referring to the text.
-Although there are many figures, the key performance data was inadequately presented.
-The novelty is not accurate at all. Why the authors prepared such Er2FeSbO7/BiTiSbO6 EBHP catalyst comparing with the reported divers different materials? Recently, the different functionalized composite materials were prepared for photo-catalytic property in different goals.
-Author should cite and discuss these article introduction section.
Besides, there are severe mistake of scientific writing in the manuscript. Author must polish the English and discuss in a logical way.
-Results and discussion: - To increase the scientific value of the manuscript Authors should consider extension of the all results section with comparison of obtained results with the results described in previous publications.
-Please identify and discuss the intermediates produced during the degradation reaction of ENR by Er2FeSbO7/BiTiSbO6 EBHP. The recycling photo-activity test should be added in the revised text.
-Structural stability of Er2FeSbO7/BiTiSbO6 EBHP is important factor for catalytic materials, so, the authors should provide XRD results to confirm the stability after recycling measurements.
-What are the final product after ENR degradation with Er2FeSbO7/BiTiSbO6 EBHP?
-The introduction section is very short and poorly described. It doesn't present the reference to the manuscript scope in most. You must be add a new sentence about nanostructure materials and advantages of them and application of them. In the introduction section, authors should make an in-depth literature review concerning the application of nanostructures in various fields. Introduction has deficiency citation to valuable works published before.
-Indeed, there are impressive amount of results. However, the conclusions section needs to improve with selected and highlighted main findings.
Reviewer 3 Report
I have read the manuscript “Synthesis, property examination of Er2FeSbO7/BiTiSbO6 heterojunction composite catalyst and photocatalytic degradation of enrofloxacin within pharmaceutical wastewater under visible light irradiation”, by Luan et al. It is one interesting study, showing the synthesis and characterization (using SEM, XRD, UV-vis and XPS) of heterojunction composite catalyst and their efficiency of photocatalytic degradation of enrofloxacin within pharmaceutical wastewater under simulated solar light. Moreover, the concentration of total organic carbon was followed and reaction intermediates formed were studied in detail. Also, reutilization study was done.
I think it is suitable for publishing in Materials.
However, there are some major revisions required to address before being considered to be accepted.
- In Introduction section, the authors should add information about which concentrations of enrofloxacin are found in the environment. Authors should also provide information on whether any photocatalytic degradation studies of this antibiotic have been performed so far.
- In section “Photocatalytic activity”: Is there any connection between the characterization results and the efficiency of photocatalytic degradation? The results should be linked and some explanation given for the existing catalyst efficiency.
- Has the adsorption efficiency of catalysts examined in the dark? If so, was the antibiotic adsorbed on the catalysts and in what percentage?
- How did you decide what concentration of radical scavengers you would use in the experiments?
- Where are the results related to section “3.7 Photoelectrochemical Experiments”?
Reviewer 4 Report
Comments
The article describes synthesis of a heterojunction composite catalyst and the photocatalytic degradation of an antibiotic from wastewater.
The theme is interesting, although some concerns must be addressed:
- The title must be corrected: the values for Er and O must be written as subscripts.
- The abstract must be restructured/reduced.
- Please remove the underline from the test throughout the manuscript (example: “For the purpose of improving the catalytic activity of… “).
- All figures must be included after their first mention in the text.
- Please provide a revised figure 2 (in the right corner of the figure there are some items that cannot be distinguished); also, the figure resolution is very poor.
- Please group the diagrams a to f from figure 5 so that they are on a single page; also, increase the fonts and the resolution so that can be readable.
- Please ensure that the captions of figures 10, 12 and 13 are on the same page with the figures.
- No information about the specific surface area and pores dimensions were provided for the prepared catalyst. It is mandatory to provide this information.
- Please provide a revised figure 18 so that the compounds names can be readable.
- Please provide full information about used reagents and instruments (company, city, country).
- Also, due to the high number of figures, please consider to introduce some of them in a supplementary material.
- It is mandatory that the English language is revised by a native English speaker.
Round 2
Reviewer 1 Report
The authors have addressed most of the comments in the revised version. The current version could be accepted for publication.
Reviewer 3 Report
Accept
Reviewer 4 Report
The manuscript can be accepted in the present form.